biomathematics/mathematical modelling/health and disease and epidemiology

vaccines, epidemic model, COVID-19 elimination, quarantine, surveillance testing

**Author for correspondence:**
Michael J. Plank
e-mail: michael.plank@canterbury.ac.nz

# Vaccination and testing of the border workforce for COVID-19 and risk of community outbreaks: a modelling study

Michael J. Plank[1,4], Rachelle N. Binny[2,4], Shaun C. Hendy[3,4], Audrey Lustig[2,4] and Kannan Ridings[3,4]

[1]School of Mathematics and Statistics, University of Canterbury, Christchurch, New Zealand
[2]Manaaki Whenua, Lincoln, New Zealand
[3]Department of Physics, University of Auckland, Auckland, New Zealand
[4]Te Pūnaha Matatini: the Centre for Complex Systems and Networks, Auckland, New Zealand

 MJP, 0000-0002-7539-3465; RNB, 0000-0002-3433-0417;
SCH, 0000-0003-3468-6517

Throughout 2020 and the first part of 2021, Australia and New Zealand have followed a COVID-19 elimination strategy. Both countries require overseas arrivals to quarantine in government-managed facilities at the border. In both countries, community outbreaks of COVID-19 have been started via infection of a border worker. This workforce is rightly being prioritized for vaccination. However, although vaccines are highly effective in preventing disease, their effectiveness in preventing infection with and transmission of SARS-CoV-2 is less certain. There is a danger that vaccination could prevent symptoms of COVID-19 but not prevent transmission. Here, we use a stochastic model of SARS-CoV-2 transmission and testing to investigate the effect that vaccination of border workers has on the risk of an outbreak in an unvaccinated community. We simulate the model starting with a single infected border worker and measure the number of people who are infected before the first case is detected by testing. We show that if a vaccine reduces transmission by 50%, vaccination of border workers increases the risk of a major outbreak from around 7% per seed case to around 9% per seed case. The lower the vaccine effectiveness against transmission, the higher the risk. The increase in risk as a result of vaccination can be mitigated by increasing the frequency of routine testing for high-exposure vaccinated groups.

# 1. Introduction

Australia and New Zealand have a strategy to eliminate community transmission of COVID-19 and to prevent reintroductions by requiring overseas arrivals to spend a period of time in government-managed isolation and quarantine facilities. Between October 2020 and March 2021, the majority of cases in both countries have been in overseas arrivals and have been contained at the border. The managed isolation and quarantine facilities and other border operations are supported by a large and varied workforce. In New Zealand in the period from 1 July 2020 to 31 March 2021, there have been 14 known border breaches resulting in an active case of COVID-19 entering the community [1]. Of these 12 breaches, six have been caused by a quarantine worker becoming infected and three have been associated with transmission between recent arrivals within quarantine facilities. The remaining five known breaches either started from a non-quarantine border worker (e.g. port worker, aircrew) or have an unknown cause. Australia experienced 12 border breaches in the same period, all associated with the infection of a border worker [2].

While there is no community transmission of SARS-CoV-2, frontline border workers are the group with the highest risk of becoming infected. Border workers and their household contacts are, therefore, rightly being prioritized for vaccination in Australia and New Zealand [3,4]. Approved vaccines are proven to be effective in preventing symptomatic disease, so vaccinating high-exposure groups will protect them from the health impacts of COVID-19. However, it remains uncertain how effective vaccines will be in reducing infection with and transmission of SARS-CoV-2. If the vaccine is effective in preventing frontline workers from becoming infected with or transmitting SARS-CoV-2, vaccination of frontline workers will provide an additional buffer that will help protect the wider community against border incursions. However, if the vaccine does not prevent infection or transmission, there is counterintuitively a danger that vaccinating frontline workers could increase the risk of community outbreaks, by making the initial infection in a frontline worker harder to detect.

In this report, we use a model for COVID-19 transmission and testing to investigate the risk of a community outbreak under various border workforce vaccination scenarios. We use the model to investigate the extent to which risks can be mitigated by routine testing. Although we apply the model to New Zealand's border workforce, the approach and conclusions are more generally applicable to situations where a particular group is vaccinated as a defence against an infectious disease entering into a larger, unvaccinated population. This includes countries using border controls to limit the introduction of an infectious disease for which a vaccine is available but not yet widely administered.

# 2. Methods

We use a stochastic branching process model for the spread of COVID-19, seeded with a single infected frontline worker. The following is a non-technical summary of the main model assumptions; see the electronic supplementary material for full model specification. The model includes heterogeneity in the number of secondary infections caused by a single infected individual (i.e. superspreading) with dispersion parameter $k = 0.5$ [5,6]. We run the model for two different values of the basic reproduction number: $R_0 = 2.5$ representing pre-existing variants of SARS-CoV-2 that were dominant for most of 2020; and $R_0 = 3.75$ representing a newer variant that is 50% more transmissible (e.g. [7]). For simplicity, we assume that frontline border workers and the general population have the same mean reproduction number and the same dispersion parameter $k = 0.5$. We assume that 33% of SARS-CoV-2 infections in unvaccinated individuals are subclinical (see the electronic supplementary material, table S3 for sensitivity analysis on this parameter), meaning that they do not develop clinical symptoms at any time during the infection. We assume subclinical infections have 50% of the transmission rate of clinical cases [8–10].

The New Zealand border workforce is heterogeneous and different groups have different frequencies of routine testing depending on their level of exposure. For simplicity, we focus on the highest-exposure group of quarantine workers who are currently on a weekly testing schedule. We assume all frontline border workers undergo regular scheduled nasopharyngeal PCR tests with symptom checks. The border worker seed case is assumed to be infected at a random time relative to their routine testing schedule. We assume that for routine tests, the probability of a positive result being returned is a function of time since infection, using the data of Kucirka *et al.* [11]. We assume that symptom checks by health professionals help to provide a very low probability that clinical cases are missed by testing after symptom onset. For example, symptomatic individuals may be retested and/or diagnosed as

probable cases in the absence of a positive PCR test result. We assume that routine tests in subclinical infections have a probability of returning a positive result that is 65% of that for clinical cases [12,13]. We assume that the time-dependence of the probability of testing positive is the same for all individuals, and that multiple tests in the same individual are statistically independent. These are model simplifications and could be generalized, for example, by explicitly modelling heterogeneity in magnitude and timing of peak viral load [14], but at the expense of increased model complexity. We assume there is no routine testing in the general population. In addition to routine testing, cases may also be detected as a result of symptom-triggered testing. When border workers are vaccinated, this only occurs in the general population. We assume that clinical cases in the general population have a 30% probability of detection by symptom-triggered testing, with a mean time from symptom onset to detection of 6 days. Neither vaccinated individuals nor subclinical infections can receive a symptom-triggered test.

We investigate how vaccinating the border workforce affects the risk of a community outbreak under various vaccine effectiveness and testing scenarios. Approved vaccines are known to be effective in preventing symptomatic disease caused by SARS-CoV-2. However, it is still uncertain how effective they are in reducing infection with or transmission of the virus. If a vaccine prevents or reduces symptoms of COVID-19 in frontline border workers but does not prevent them from transmitting the virus, there is counterintuitively a danger this could increase the risk of community outbreaks by making it harder to detect the virus in the seed cases, and therefore more likely that the outbreak could spread into the community before being detected.

We assume that 100% of the frontline border workforce has been fully vaccinated, and that the vaccine is 100% effective in preventing symptoms of COVID-19. This is a simplifying assumption, but it is a conservative one for the purposes of this study, which is to estimate the risk of asymptomatic transmission from border workers. Early uptake of the vaccine among New Zealand's border workers has been high, with approximately 91% having received their first dose as of 17 March 2021 [15]. The Pfizer/BioNTech mRNA vaccine being rolled out in New Zealand has been shown to have high effectiveness (greater than 90%) against symptoms [16,17] so the assumption, although conservative, is not unreasonable. A sensitivity analysis where the vaccine is 80% effective in preventing symptoms in breakthrough infections is provided in the electronic supplementary material.

We first investigate a worst-case scenario in which the vaccine does not reduce infection or transmission of the virus at all. This scenario is not intended to be a realistic vaccine model, but rather to illustrate the nature of the risks involved, and their potential magnitude in a worst-case scenario. In this scenario, all infected individuals transmit the virus at the same rate as if they were unvaccinated. We then investigate more realistic scenarios in which the vaccine is partially effective in preventing infection and transmission. To model this, we assume that the vaccine has probability $VE_I$ of preventing infection and reduces transmission in breakthrough infections to $(1 - VE_T)/(1 - VE_I)$ relative to an unvaccinated individual (see the electronic supplementary material for more details). In this parametrization, $VE_T$ is the overall effectiveness against transmission as a combination of (i) infection prevention and (ii) transmission reduction in breakthrough infections. We investigate scenarios where $VE_T$ is 50% or 75% and where $VE_I$ is either zero (meaning no infection prevention) or 50% of $VE_T$ (meaning that half of the overall transmission reduction effect comes from infection prevention). This is approximately consistent with studies showing a 50% reduction in secondary attack rates among household members of vaccinated cases [18].

Prevention of symptomatic disease in vaccinated border workers has two effects on the testing model. Firstly, it means that border workers only get routine scheduled tests (e.g. weekly tests) and do not get additional symptom-triggered tests that they may get if they were unvaccinated. Secondly, it means that cases cannot be flagged for repeat testing or diagnosed as a probable case as a result of symptom checks. This increases the likelihood of an infected individual being missed by a PCR test.

Finally, we consider a scenario modelling vaccination of frontline workers and their close family members. To model this, we assume that 50% of the cases infected by a frontline worker are vaccinated. This is a simplifying assumption designed to give an indication of the qualitative effect of vaccinating frontline workers' family members. A more detailed model stratified into the household and non-household contacts could be used to provide a more accurate result.

For each scenario, we examine the risk of community outbreaks under different routine testing frequencies for border workers. We calculate the proportion of simulations, each seeded with a single infected border worker, in which the outbreak is (i) never detected, (ii) first detected in the seed case (generation 1 detection), or (iii) first detected in a secondary case or later (generation 2+ detection). We also calculate the size of the outbreak (total number of people infected) at the time it is first detected.

# 3. Results

Table 1 shows the proportion of 5000 independent realizations initialized with a single infected frontline border worker in which the outbreak: (i) dies out without being detected, (ii) is detected in generation 1 (i.e. in the seed case representing the border worker), or (iii) is detected in generation 2 or later (i.e. a non-seed case), with $R_0 = 2.5$. It also shows the proportion of simulations in which 10 people or 40 people are infected before the outbreak is detected. Note we do not present confidence intervals for these probabilities as these would account only for the relative small amount of stochasticity in a multinomial random variable over 5000 trials and would ignore other sources of model uncertainty.

Undetected outbreaks typically occur when the seed case does not infect any other individuals, or only causes a very small number of additional infections. These outbreaks, therefore, do not represent a significant public health risk. For outbreaks detected in generation 1 or in generation 2+, table 1 shows the median number of infected individuals at the time of detection (referred to as outbreak size). There is some variability between simulations of the model, due to the timing of the seed cases becoming infected relative to their routine testing schedule, as well as the inherent stochasticity in the testing and transmission models. To capture this variability, table 1 also shows the interquartile range (IQR) of outbreak size.

In all scenarios, outbreaks detected at generation 2+ are significantly larger than those detected at generation 1, which typically involve fewer than five infections at the time of detection. Consistent with previous modelling studies [19], this shows that if a case is detected outside the frontline worker group, it is likely that there is a much larger number of people already infected. This provides an early indicator of the potential size of an outbreak, enabling a timely response in the absence of direct information on the number of infections [19,20]. We, therefore, focus on the proportion of generation 2+ detections as a key model output quantifying the risk of a serious community outbreak. The hitting probabilities for 10 infections and 40 infections, and the average size of generation 2+ detections also indicate the potential size of these outbreaks.

In a baseline scenario with no vaccination and with scheduled weekly testing of frontline workers (representing the current situation in early February 2021), approximately 7% of simulations result in an outbreak detected at generation 2+ and these have median outbreak size 18.5 with IQR [8, 41]. In the worst-case scenario of a vaccine that does not reduce transmission, and with scheduled weekly testing, approximately 14% of simulations result in an outbreak detected at generation 2+ and these have a median outbreak size of 23 [10, 46]. This shows that in a worst-case scenario, vaccinating border workers could approximately double the frequency of generation 2+ detections and slightly increase the average outbreak size. It also more than doubles the risk of an outbreak infecting 40 people before being detected. If the frequency of routine testing is increased to once every 4 days, the key metrics (probability of and outbreak size for generation 2+ detections, and probability of reaching 40 infections) are close to their baseline values. This shows that in the worst-case scenario, the risk of community outbreaks due to vaccine-induced symptom prevention in frontline workers can be completely mitigated by increasing the testing frequency from weekly to once every 4 days.

A more realistic scenario is one where the vaccine provides at least some reduction in transmission, although the size of the reduction and relative contribution of infection prevention and transmission reduction in breakthrough infections are uncertain at this time. If the vaccine reduces transmission by 50% with no infection prevention (denoted VE 0/50% in table 1), the risk of community outbreaks is not as great as when the vaccine does not reduce transmission. With weekly testing, the probability of a generation 2+ detection is around 9%, compared to 7% in the baseline no-vaccine scenario and the probability of reaching 40 infections is around 3% compared to 2.5% at baseline. Increasing the testing frequency from once every 7 days to once every 4 days more than compensates for this increased risk, reducing the probability of a generation 2+ detection to around 3.4% and the probability of reaching 40 infections to 1.1%, which are lower than in the baseline scenario. Risks are similar although slightly lower when half of the overall effect of the vaccine comes from infection prevention (VE 25/50% in table 1). Therefore, with a vaccine that is only partially effective against transmission, increased testing of frontline workers is needed in order to avoid increasing the risk of a serious community outbreak.

Under a scenario where the vaccine is 75% effective in reducing transmission, whether entirely from reducing transmission from breakthrough infections (VE 0/75% in table 1) or from a combination of infection prevention and transmission reduction (VE 38/75% in table 1), the key risk metrics are always lower than in the baseline scenario with no vaccination. This shows that a vaccine that is highly effective in reducing spread, maintaining a testing frequency of once per week is sufficient.

Table 2 shows corresponding results for a SARS-CoV-2 variant that is 50% more transmissible ($R_0 = 3.75$). This increases the probability of generation 2+ detections and results in larger outbreaks

**Table 1.** Model results for four vaccination scenarios (no vaccination of frontline workers and different levels of vaccine effectiveness) and different intervals for routine testing of frontline workers, with $R_0 = 2.5$. Vaccine effectiveness (VE) is stated as two values, the first is the effectiveness against infection (VE$_I$) and the second is the overall effectiveness against transmission (VE$_T$) as a combination of infection prevention and reduced transmission in breakthrough infections. Row with italic font indicates the status quo prior to vaccination of border workers. The 'detection type' columns show the proportion of model simulations that result in: an outbreak that dies out without being detected (Undet.); an outbreak that is first detected in the seed case (Gen. 1); and an outbreak that is first detected in a secondary case or later (Gen. 2+). The 'hitting probability' columns show the proportion of simulations in which the outbreak reaches at least 10 infected individuals (P10) or at least 40 infected individuals (P40) before the first case is detected. The 'outbreak size' columns show the median [interquartile range (IQR)] number of infected individuals at the time the first case is detected. Results are from 5000 independent simulations of the model, each initialized with a single seed case in a frontline worker.

| scenario | test interval (days) | detection type | | | hitting probabilities | | outbreak size | |
|---|---|---|---|---|---|---|---|---|
| | | Undet. (%) | Gen. 1 (%) | Gen. 2+ (%) | P10 (%) | P40 (%) | Gen. 1 | Gen. 2+ |
| *no vaccine* | *7* | *6.0* | *87.3* | *6.6* | *14.2* | *2.5* | *1 [1, 4]* | *18.5 [8, 41]* |
| no vaccine | 4 | 2.5 | 93.7 | 3.9 | 12.0 | 1.4 | 1 [1, 4] | 15 [7, 31] |
| no vaccine | 2 | 0.4 | 98.3 | 1.3 | 9.1 | 0.5 | 1 [1, 4] | 12 [5.5, 20] |
| VE 0% | 7 | 9.2 | 77.0 | 13.8 | 24.6 | 6.2 | 2 [1, 6] | 23 [10, 46] |
| VE 0% | 4 | 2.6 | 91.4 | 6.0 | 18.5 | 3.2 | 2 [1, 6] | 21.5 [9, 43] |
| VE 0% | 2 | 0.2 | 98.0 | 1.7 | 11.0 | 0.8 | 2 [1, 4] | 10 [5, 21.3] |
| VE 0/50% | 7 | 11.6 | 79.8 | 8.6 | 13.7 | 3.0 | 1 [1, 3] | 16 [7, 38] |
| VE 0/50% | 4 | 3.5 | 93.0 | 3.4 | 9.0 | 1.1 | 1 [1, 3] | 15.5 [6, 33] |
| VE 0/50% | 2 | 0.3 | 98.8 | 0.9 | 4.4 | 0.3 | 1 [1, 3] | 8 [5, 26.5] |
| VE 25/50% | 7 | 32.8 | 59.3 | 7.9 | 12.8 | 2.8 | 1 [1, 4] | 20 [7, 40] |
| VE 25/50% | 4 | 27.3 | 69.2 | 3.5 | 9.2 | 1.1 | 1 [1, 4] | 13 [6, 32.8] |
| VE 25/50% | 2 | 25.2 | 73.9 | 0.9 | 5.9 | 0.4 | 1 [1, 3] | 9 [4, 18] |
| VE 0/75% | 7 | 12.5 | 82.0 | 5.5 | 7.5 | 1.6 | 1 [1, 2] | 16 [5.3, 36] |
| VE 0/75% | 4 | 4.1 | 93.9 | 2.0 | 4.0 | 0.6 | 1 [1, 2] | 10 [5, 31] |
| VE 0/75% | 2 | 0.4 | 99.1 | 0.5 | 2.0 | 0.1 | 1 [1, 2] | 6 [3, 19] |
| VE 38/75% | 7 | 44.9 | 50.9 | 4.1 | 6.8 | 1.4 | 1 [1, 3] | 17.5 [8, 37] |
| VE 38/75% | 4 | 39.7 | 58.1 | 2.2 | 4.6 | 0.7 | 1 [1, 2] | 11 [5, 28] |
| VE 38/75% | 2 | 37.7 | 61.9 | 0.4 | 2.3 | 0.1 | 1 [1, 2] | 8 [3, 13] |

**Table 2.** Model results for four vaccination scenarios (no vaccination of frontline workers and different levels of vaccine effectiveness) and different intervals for routine testing of frontline workers, with $R_0 = 3.75$. Results are from 5000 independent simulations of the model, each initialized with a single seed case in a frontline worker.

| scenario | test interval (days) | detection type | | | hitting probabilities | | outbreak size | |
|---|---|---|---|---|---|---|---|---|
| | | Undet. (%) | Gen. 1 (%) | Gen 2+ (%) | P10 (%) | P40 (%) | Gen. 1 | Gen. 2+ |
| no vaccine | 7 | 4.8 | 86.9 | 8.3 | 23.4 | 6.6 | 2 [1, 7] | 30 [14, 69] |
| no vaccine | 4 | 2.2 | 92.5 | 5.4 | 20.8 | 5.0 | 2 [1, 6] | 31 [12, 73.5] |
| no vaccine | 2 | 0.3 | 97.9 | 1.9 | 15.8 | 1.8 | 2 [1, 6] | 15 [7, 31] |
| VE 0% | 7 | 7.4 | 75.5 | 17.1 | 34.8 | 13.8 | 2 [1, 10] | 41 [16, 82] |
| VE 0% | 4 | 2.5 | 88.9 | 8.6 | 28.4 | 8.5 | 2 [1, 9] | 32 [16, 70] |
| VE 0% | 2 | 0.3 | 97.4 | 2.4 | 21.6 | 3.5 | 2 [1, 7] | 16 [8, 39] |
| VE 0/50% | 7 | 9.1 | 79.5 | 11.4 | 22.4 | 7.6 | 1 [1, 6] | 30 [11, 62.5] |
| VE 0/50% | 4 | 2.9 | 92.2 | 4.9 | 17.6 | 4.2 | 1 [1, 5] | 26 [10.5, 55] |
| VE 0/50% | 2 | 0.3 | 98.2 | 1.5 | 10.6 | 1.0 | 1 [1, 4] | 12 [6, 26.8] |
| VE 25/50% | 7 | 31.6 | 57.9 | 10.4 | 21.2 | 8.0 | 2 [1, 7] | 36 [13, 76] |
| VE 25/50% | 4 | 27.1 | 68.2 | 4.7 | 16.2 | 3.8 | 2 [1, 6] | 23 [11, 60] |
| VE 25/50% | 2 | 25.2 | 73.5 | 1.3 | 11.0 | 1.1 | 2 [1, 5] | 24 [11.3, 37] |
| VE 0/75% | 7 | 11.7 | 80.2 | 8.1 | 13.8 | 4.2 | 1 [1, 3] | 25 [8, 59] |
| VE 0/75% | 4 | 3.4 | 93.5 | 3.2 | 9.2 | 1.8 | 1 [1, 2] | 18 [6, 47] |
| VE 0/75% | 2 | 0.4 | 98.7 | 0.9 | 5.1 | 0.4 | 1 [1, 2] | 12 [5.5, 29] |
| VE 38/75% | 7 | 43.6 | 49.9 | 6.5 | 12.6 | 3.9 | 1 [1, 5] | 27.5 [11, 63] |
| VE 38/75% | 4 | 39.6 | 57.8 | 2.6 | 8.0 | 1.7 | 1 [1, 3] | 18 [9.8, 48.3] |
| VE 38/75% | 2 | 37.7 | 61.5 | 0.8 | 5.5 | 0.6 | 1 [1, 3] | 12.5 [4, 27] |

across all scenarios investigated. As for the original, less transmissible variant, a vaccine that reduces transmission by 50% requires an increase in testing frequency to avoid increasing the risk of generation 2+ detections or of outbreaks reaching 40 infections relative to the baseline no-vaccine scenario. Testing border workers once every 4 days is sufficient to reduce the risk of community outbreaks from the more transmissible variant to a level similar to or below that from the original variant. In the more effective vaccine scenarios (75% transmission reduction), the risk of community outbreaks with weekly testing is similar to or slightly below the risk with no vaccination. Increasing the testing frequency to once every 4 days reduces risk below the level for the original variant.

Table 3 shows the results of the model for vaccinating frontline workers and their close family members. Overall, the results are similar to those where only frontline workers are vaccinated (table 1). In the worst-case scenario where the vaccine does not reduce transmission, there is a slightly lower probability of a generation 2+ detection when close family members are vaccinated (11%) than when they are not vaccinated (14%). However, there is a slightly higher median outbreak size for generation 2+ detections (27 compared to 23) and a very similar probability of an outbreak that reaches 40 infections before detection (6.6% compared to 6.2%). The risk of large outbreaks is increased because family members do not get routine testing and, if they are vaccinated, will not develop symptoms. There is, therefore, a danger of them continuing to transmit the virus asymptomatically without detection, allowing the outbreak to grow larger before it is eventually picked up in a symptomatic case. If the vaccine is at least 50% effective against transmission, this danger is much smaller and the model outputs are closer to the scenario where family members are not vaccinated (table 1).

## 4. Discussion

We used a stochastic model of SARS-CoV-2 transmission and testing to assess the risk of community outbreaks under various frontline worker vaccination and testing scenarios. Preliminary results for the effectiveness of the Pfizer vaccine being rolled out to New Zealand's border workforce on transmission are promising. There is preliminary data showing that the Pfizer vaccine reduces viral load in infected individuals [21,22], reduces the incidence of document symptomatic or asymptomatic infection [23] and reduces transmission to close contacts [24]. There are also encouraging results from other mRNA vaccines that vaccination reduces asymptomatic infection [23,25]. Together, this evidence suggests that vaccination will reduce transmission substantially. However, until more data is available and studies have been through the peer review process, a precautionary approach is warranted and this means planning for a scenario where the vaccine has relatively low effectiveness at preventing infection or reducing transmission.

Under a worst-case scenario of a vaccine that prevents symptoms but does not reduce infection or transmission at all, vaccination of frontline workers could approximately double the risk of a large community outbreak. This risk could be mitigated by increasing the frequency of routine testing of frontline workers from once per week to once every 4 days. Under a more realistic scenario of a vaccine that is 50% effective against transmission, the increase in risk due to vaccination of frontline workers is smaller, but increased testing frequency is still required to avoid increasing the risk. Under a scenario where the vaccine is 75% effective against transmission, vaccination of border workers reduces the risk of community outbreaks without the need for increased testing frequency.

Some of the recently identified variants of SARS-CoV-2 are thought to be more transmissible. For example, the B.1.1.7 variant first identified in the UK has been estimated to have a 43–90% higher reproduction number than pre-existing variants [7]. Contact tracing data from England suggested B.1.1.7 has a 10–70% higher secondary attack rate [26]. We reran our model with a 50% increase in transmissibility ($R_0 = 3.75$). Overall, this increases the expected frequency and size of community outbreaks across all scenarios. Testing frontline border workers every 4 days is sufficient to mitigate this risk in all but the worst-case scenario of a vaccine that does not reduce transmission at all.

New Zealand's vaccine rollout has initially prioritized frontline border workers and their close family members. Since family members are not required to undergo routine regular testing, there is a danger that a vaccine that prevented symptoms could turn family members into asymptomatic spreaders. A model scenario where 50% of secondary cases from frontline workers were vaccinated showed that this could potentially increase the risk of large community outbreaks. This suggests that careful attention should be paid to any groups who are vaccinated because of their proximity to groups like

**Table 3.** Model for vaccination of frontline workers and their close family members under four vaccination scenarios (no vaccination and different levels of vaccine effectiveness) and different intervals for routine testing of frontline workers, with $R_0 = 2.5$. Results are from 5000 independent simulations of the model, each initialized with a single seed case in a frontline worker.

| scenario | test interval (days) | detection type | | | hitting probabilities | | outbreak size | |
|---|---|---|---|---|---|---|---|---|
| | | Undet. (%) | Gen. 1 (%) | Gen 2+ (%) | P10 (%) | P40 (%) | Gen. 1 | Gen. 2+ |
| no vaccine | 7 | 6.2 | 88.4 | 5.4 | 15.1 | 2.8 | 1 [1, 4] | 26 [12.8, 53] |
| no vaccine | 4 | 2.3 | 94.9 | 2.8 | 11.7 | 1.5 | 1 [1, 4] | 23 [10, 42.3] |
| no vaccine | 2 | 0.2 | 99.0 | 0.8 | 9.1 | 0.6 | 1 [1, 4] | 18 [7, 34] |
| VE 0% | 7 | 9.1 | 80.0 | 10.9 | 24.8 | 6.6 | 2 [1, 7] | 27 [12, 57] |
| VE 0% | 4 | 2.8 | 92.5 | 4.7 | 18.6 | 3.3 | 2 [1, 6] | 26 [11, 48] |
| VE 0% | 2 | 0.2 | 98.6 | 1.2 | 12.7 | 1.0 | 2 [1, 5] | 12 [7.8, 38] |
| VE 0/50% | 7 | 11.5 | 81.9 | 6.7 | 12.5 | 2.6 | 1 [1, 3] | 23 [9, 40] |
| VE 0/50% | 4 | 3.6 | 93.7 | 2.7 | 7.6 | 1.3 | 1 [1, 3] | 16 [7, 40] |
| VE 0/50% | 2 | 0.5 | 98.9 | 0.6 | 4.4 | 0.1 | 1 [1, 2] | 10 [7, 14] |
| VE 25/50% | 7 | 32.7 | 61.1 | 6.2 | 11.7 | 2.5 | 1 [1, 4] | 21 [11.8, 44.3] |
| VE 25/50% | 4 | 27.4 | 70.2 | 2.5 | 6.8 | 0.8 | 1 [1, 4] | 15 [6, 34.8] |
| VE 25/50% | 2 | 25.2 | 74.2 | 0.6 | 4.4 | 0.3 | 1 [1, 3] | 7 [4, 15.8] |
| VE 0/75% | 7 | 12.6 | 83.9 | 3.5 | 5.2 | 1.2 | 1 [1, 2] | 17 [7, 43] |
| VE 0/75% | 4 | 4.1 | 94.7 | 1.3 | 3.1 | 0.4 | 1 [1, 2] | 15 [6.3, 31] |
| VE 0/75% | 2 | 0.5 | 99.1 | 0.4 | 1.3 | 0.1 | 1 [1, 2] | 4 [3, 24.8] |
| VE 38/75% | 7 | 44.6 | 52.6 | 2.8 | 4.6 | 0.9 | 1 [1, 2] | 15 [7, 34] |
| VE 38/75% | 4 | 39.7 | 58.9 | 1.4 | 2.9 | 0.5 | 1 [1, 2] | 12.5 [5, 40] |
| VE 38/75% | 2 | 37.7 | 62.1 | 0.2 | 1.2 | 0.0 | 1 [1, 2] | 6.5 [4, 11] |

border workers with high levels of exposure to the virus, but who are not themselves undergoing routine testing. Again, the risk can be mitigated by increasing the testing frequency for frontline workers.

For simplicity, we assumed that all tests have the same time-dependent sensitivity curve [11], representing a PCR nasopharyngeal swab test. Increasing the frequency of nasopharyngeal swabbing of frontline border workers may not be practical and would be onerous for those being tested. Regular saliva testing or rapid antigen testing in combination with a weekly nasopharyngeal swab test may be an alternative method to achieve the routine testing coverage needed to mitigate the risk of a community outbreak. A review found that a range of saliva tests had comparable sensitivity and specificity to nasopharyngeal swabbing [27]. Even if saliva tests had lower sensitivity, this could be compensated for by increased testing frequency provided sufficient processing capacity was available.

Rolling out a rapid mass vaccination programme to as many people as possible is the best way to prevent the potentially devastating health impacts of COVID-19. Countries such as New Zealand, Australia and Taiwan have largely eliminated community transmission of the virus and have a strategy of mandatory quarantine for incoming travellers to prevent re-establishment [28]. In such countries, people working in border quarantine facilities have the highest exposure to the virus of any population group. Our results should not be used to suggest that border workers should not be vaccinated as a priority. Protecting them from the health impacts of COVID-19 at the earliest opportunity is an ethical obligation. However, care needs to be taken to ensure this group does not inadvertently become a silent source of transmission into a community that is largely unvaccinated at present. Regular routine testing is a good safeguard against this. If, in future, vaccines are demonstrated to be highly effective against transmission, the danger will be much smaller. Until then, or until high levels of vaccine coverage are achieved in the general population, increased routine testing of border workers is recommended.

Data accessibility. Code for running the model is provided as electronic supplementary material.

Authors' contributions. M.J.P. conceived the study, designed the model, carried out the numerical simulations and drafted the manuscript. R.N.B. designed the model and critically revised the manuscript. S.C.H. designed the model and critically revised the manuscript. A.L. designed the model and critically revised the manuscript. K.R. designed the model and critically revised the manuscript.

Competing interests. We declare we have no competing interests.

Funding. This work was funded by the New Zealand Ministry of Business, Innovation and Employment and Te Pūnaha Matatini, Centre of Research Excellence in Complex Systems.

Acknowledgements. The authors acknowledge the support of Stats NZ, ESR and the New Zealand Ministry of Health in supplying data in support of this work. We are grateful to Samik Datta, Nigel French, Ricci Harris, Markus Luczak-Roesh, Fraser Morgan, Matt Parry, Patricia Priest, Ian Town and members of the Vaccine Modelling Group chaired by Therese Egan for feedback and informal peer review comments on an earlier version of this manuscript.

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
