## [Peer Review File · Royal Society Open Science]

Review History

RSOS-210686.R0 (Original submission)

Review form: Reviewer 1

Is the manuscript scientifically sound in its present form?

Yes

Are the interpretations and conclusions justified by the results?

No

Is the language acceptable?

Yes

Do you have any ethical concerns with this paper?

No

Have you any concerns about statistical analyses in this paper?

No

Recommendation?

Major revision is needed (please make suggestions in comments)

Comments to the Author(s)

This paper presents a scenario analysis on the risk of community outbreaks of COVID-19 associated with vaccinating border workers in a largely unvaccinated population. It asks an important question as to whether this approach actually reduces risk in populations with an elimination strategy given differing reproduction numbers and vaccine efficacy against transmission. Although this paper presents an interesting idea and model to answer an important question, it is my opinion that it overlooks some very important impacts of vaccination, which makes it difficult to interpret how this analysis relates to risk of community outbreaks. In particular, the effect of vaccination on susceptibility to infection is overlooked. I believe this paper has a lot of potential, but would be inappropriate to publish until it addresses some of my concerns (given below).

Main points:

The paper only makes mention of the effect of vaccines on transmission but does not note its effect on susceptibility to infection. This paper considers a model that is initialised by seeding a case in a border worker, but this is not put in context of a lower seeding rate due to vaccination.

On a related note, their third scenario simulates transmission in a context where some contacts of border workers are vaccinated. The effect of vaccine efficacy against infection is important here as it impacts both the seeding rate and the outbreak dynamics.

Assuming 100% vaccine efficacy against symptoms is noted as being conservative and it is partially justified based on estimates of >90% efficacy against symptomatic infection. I think this overlooks the fact that conditional upon being infected the probability of symptoms may still be rather high. For example, suppose that the vaccine was 90% effective against infections and 90% effective against symptomatic infections, the probability that an infected worker would be symptomatic is still 33%. In the absence of reasonable estimates of efficacy against symptomatic infection I think this warrants a sensitivity analysis.

Minor points:

There's no reference to the dispersion parameter $k=0.5$, I've seen estimates as low as $k=0.1$. It would either be worth knowing where this parameter came from or to see a sensitivity analysis with respect to dispersion.

Similar to above, I would guess that results are quite sensitive to the transmission rate in subclinical infections.

The output in terms of probability of detection at different generations as well as the different generation sizes seems like an unintuitive proxy for risk. I found it not entirely straight forward to weigh up detection probabilities with the outbreak sizes when comparing the different scenarios. Maybe a more straightforward measure of risk would be the epidemic size distribution at the time of detection, or even hitting probabilities of epidemic thresholds before detection.

Review form: Reviewer 2

Is the manuscript scientifically sound in its present form?

No

Are the interpretations and conclusions justified by the results?

No

Is the language acceptable?

Yes

Do you have any ethical concerns with this paper?

Yes

Have you any concerns about statistical analyses in this paper?

No

Recommendation?

Reject

Comments to the Author(s)

The paper dose not have any mathematical model

Decision letter (RSOS-210686.R0)

Dear Dr Plank

The Editors assigned to your paper RSOS-210686 "Vaccination and testing of the border workforce for COVID-19 and risk of community outbreaks: a modelling study" have now received comments from reviewers and would like you to revise the paper in accordance with the reviewer comments and any comments from the Editors. Please note this decision does not guarantee eventual acceptance.

Please submit your revised manuscript and required files (see below) no later than 21 days from today's (ie 10-Sep-2021) date. Note: the ScholarOne system will 'lock' if submission of the revision is attempted 21 or more days after the deadline. If you do not think you will be able to meet this deadline please contact the editorial office immediately.

on behalf of Professor Christine Currie (Associate Editor) and Mark Chaplain (Subject Editor)
openscience@royalsociety.org

Associate Editor Comments to Author (Professor Christine Currie):

Associate Editor: 1

Comments to the Author:

The referee has provided some useful comments on your paper and some sensible suggestions for improving it. I agree with the referee that your paper tackles an interesting problem and consequently we are keen to see a revision of the work. Nonetheless, there are some important points that need clarifying, and some extra thought might be needed as to the assumptions made in the modelling.

Associate Editor: 2

Comments to the Author:

This is an interesting article, which uses a stochastic branching model to estimate the development of outbreaks of covid-19 caused by border control staff becoming infected and taking the infection out into the wider community. It could be argued that the work has limited applicability outside of New Zealand and Australia as much of the rest of the world still has quite significant community transmission. Many countries also have a higher level of vaccination within the general population. If it is possible to make the work more generally applicable, that might increase its audience.

Reviewer comments to Author:

Reviewer: 1

Comments to the Author(s)

This paper presents a scenario analysis on the risk of community outbreaks of COVID-19 associated with vaccinating border workers in a largely unvaccinated population. It asks an important question as to whether this approach actually reduces risk in populations with an elimination strategy given differing reproduction numbers and vaccine efficacy against transmission. Although this paper presents an interesting idea and model to answer an important question, it is my opinion that it overlooks some very important impacts of vaccination, which makes it difficult to interpret how this analysis relates to risk of community outbreaks. In particular, the effect of vaccination on susceptibility to infection is overlooked. I believe this paper has a lot of potential, but would be inappropriate to publish until it addresses some of my concerns (given below).

Main points:

The paper only makes mention of the effect of vaccines on transmission but does not note its effect on susceptibility to infection. This paper considers a model that is initialised by seeding a case in a border worker, but this is not put in context of a lower seeding rate due to vaccination.

On a related note, their third scenario simulates transmission in a context where some contacts of border workers are vaccinated. The effect of vaccine efficacy against infection is important here as it impacts both the seeding rate and the outbreak dynamics.

Assuming 100% vaccine efficacy against symptoms is noted as being conservative and it is partially justified based on estimates of >90% efficacy against symptomatic infection. I think this overlooks the fact that conditional upon being infected the probability of symptoms may still be rather high. For example, suppose that the vaccine was 90% effective against infections and 90% effective against symptomatic infections, the probability that an infected worker would be symptomatic is still 33%. In the absence of reasonable estimates of efficacy against symptomatic infection I think this warrants a sensitivity analysis.

Minor points:

There's no reference to the dispersion parameter $k=0.5$, I've seen estimates as low as $k=0.1$. It would either be worth knowing where this parameter came from or to see a sensitivity analysis with respect to dispersion.

Similar to above, I would guess that results are quite sensitive to the transmission rate in subclinical infections.

The output in terms of probability of detection at different generations as well as the different generation sizes seems like an unintuitive proxy for risk. I found it not entirely straight forward to weigh up detection probabilities with the outbreak sizes when comparing the different scenarios. Maybe a more straightforward measure of risk would be the epidemic size distribution at the time of detection, or even hitting probabilities of epidemic thresholds before detection.

===PREPARING YOUR MANUSCRIPT===

If you have been asked to revise the written English in your submission as a condition of publication, you must do so, and you are expected to provide evidence that you have received language editing support. The journal would prefer that you use a professional language editing service and provide a certificate of editing, but a signed letter from a colleague who is a native

speaker of English is acceptable. Note the journal has arranged a number of discounts for authors using professional language editing services (<https://royalsociety.org/journals/authors/benefits/language-editing/>).

===PREPARING YOUR REVISION IN SCHOLARONE===

<https://royalsociety.org/journals/authors/author-guidelines/#supplementary-material> to include a suitable title and informative caption. An example of appropriate titling and captioning may be found at https://figshare.com/articles/Table_S2_from_Is_there_a_trade-

off_between_peak_performance_and_performance_breadth_across_temperatures_for_aerobic_sc
ope_in_teleost_fishes_/3843624.

Author's Response to Decision Letter for (RSOS-210686.R0)

See Appendix A.

Decision letter (RSOS-210686.R1)

Dear Dr Plank,

It is a pleasure to accept your manuscript entitled "Vaccination and testing of the border workforce for COVID-19 and risk of community outbreaks: a modelling study" in its current form for publication in Royal Society Open Science. The comments of the reviewer(s) who reviewed your manuscript are included at the foot of this letter.

COVID-19 rapid publication process:

We are taking steps to expedite the publication of research relevant to the pandemic. If you wish, you can opt to have your paper published as soon as it is ready, rather than waiting for it to be published the scheduled Wednesday.

This means your paper will not be included in the weekly media round-up which the Society sends to journalists ahead of publication. However, it will still appear in the COVID-19 Publishing Collection which journalists will be directed to each week (<https://royalsocietypublishing.org/topic/special-collections/novel-coronavirus-outbreak>).

If you wish to have your paper considered for immediate publication, or to discuss further, please notify openscience_proofs@royalsociety.org and press@royalsociety.org when you respond to this email.

You can expect to receive a proof of your article in the near future. Please contact the editorial office (openscience@royalsociety.org) and the production office (openscience_proofs@royalsociety.org) to let us know if you are likely to be away from e-mail

contact -- if you are going to be away, please nominate a co-author (if available) to manage the proofing process, and ensure they are copied into your email to the journal.

on behalf of Professor Christine Currie (Associate Editor) and Mark Chaplain (Subject Editor)
openscience@royalsociety.org

Associate Editor Comments to Author (Professor Christine Currie):

Comments to the Author:

Many thanks for carrying out the suggested changes and providing a clear response to the referees. This is an interesting paper based on a thorough analysis.

Appendix A

Author responses (in bold) to Associate Editor and Reviewer comments

Associate Editor Comments to Author (Professor Christine Currie):

Associate Editor: 1

Comments to the Author:

The referee has provided some useful comments on your paper and some sensible suggestions for improving it. I agree with the referee that your paper tackles an interesting problem and consequently we are keen to see a revision of the work. Nonetheless, there are some important points that need clarifying, and some extra thought might be needed as to the assumptions made in the modelling.

- **Thank you for these positive remarks. We have addressed the specific points of Associate Editor 2 and Reviewer 1 – see below.**

Associate Editor: 2

Comments to the Author:

This is an interesting article, which uses a stochastic branching model to estimate the development of outbreaks of covid-19 caused by border control staff becoming infected and taking the infection out into the wider community. It could be argued that the work has limited applicability outside of New Zealand and Australia as much of the rest of the world still has quite significant community transmission. Many countries also have a higher level of vaccination within the general population. If it is possible to make the work more generally applicable, that might increase its audience.

- **We have added some commentary to the last paragraph of the Introduction to make clear the broader applicability of our approach and conclusions.**

Reviewer comments to Author:

Reviewer: 1

Comments to the Author(s)

This paper presents a scenario analysis on the risk of community outbreaks of COVID-19 associated with vaccinating border workers in a largely unvaccinated population. It asks an important question as to whether this approach actually reduces risk in populations with an elimination strategy given differing reproduction numbers and vaccine efficacy against transmission. Although this paper presents an interesting idea and model to answer an important question, it is my opinion that it overlooks some very important impacts of vaccination, which makes it difficult to interpret how this analysis relates to risk of community outbreaks. In particular, the effect of vaccination on susceptibility to infection is overlooked. I believe this paper has a lot of potential, but would be inappropriate to publish until it addresses some of my concerns (given below).

- **Thank you for these encouraging remarks. These comments are addressed in detail below.**

Main points:

The paper only makes mention of the effect of vaccines on transmission but does not note its effect on susceptibility to infection. This paper considers a model that is initialised by seeding a case in a border worker, but this is not put in context of a lower seeding rate due to vaccination.

- **This is a good point and we have now added and discussed new results for additional scenarios where the vaccine's effect on transmission comes partly from infection prevention and partly from reduction of transmission in breakthrough infections (see new text in Methods section and Tables 1-3).**

On a related note, their third scenario simulates transmission in a context where some contacts of border workers are vaccinated. The effect of vaccine efficacy against infection is important here as it impacts both the seeding rate and the outbreak dynamics.

- **We agree and this scenario now also includes results for cases where part of the vaccine's reduction in transmission comes from infection prevention (Table 3).**

Assuming 100% vaccine efficacy against symptoms is noted as being conservative and it is partially justified based on estimates of >90% efficacy against symptomatic infection. I think this overlooks the fact that conditional upon being infected the probability of symptoms may still be rather high. For example, suppose that the vaccine was 90% effective against infections and 90% effective against symptomatic infections, the probability that an infected worker would be symptomatic is still 33%. In the absence of reasonable estimates of efficacy against symptomatic infection I think this warrants a sensitivity analysis.

- **We have added a new sensitivity analysis showing the results for a scenario where the vaccine is only 80% effective in preventing symptomatic COVID-19 in breakthrough infections (Table S4).**

Minor points:

There's no reference to the dispersion parameter $k=0.5$, I've seen estimates as low as $k=0.1$. It would either be worth knowing where this parameter came from or to see a sensitivity analysis with respect to dispersion.

- **We have added supporting references for this parameter value.**

Similar to above, I would guess that results are quite sensitive to the transmission rate in subclinical infections.

- **Rather than run an additional sensitivity analysis, we have added better justification and references for the choice of this parameter value. Note that the Supplementary Material includes a sensitivity analysis to the proportion of infections that are subclinical.**

The output in terms of probability of detection at different generations as well as the different generation sizes seems like an unintuitive proxy for risk. I found it not entirely straight forward to weigh up detection probabilities with the outbreak sizes when comparing the different scenarios.

Maybe a more straightforward measure of risk would be the epidemic size distribution at the time of detection, or even hitting probabilities of epidemic thresholds before detection.

- **We agree the results provided did not enable straightforward interpretation. As suggested we have added hitting probabilities of 10 infections and 40 infections before detection to Tables 1-3 and referred to these in the Results section. The overall outbreak size at detection does not provide useful extra information because it tends to have the same median and IQR as the subset of outbreaks detected at generation 1 (penultimate column of Tables 1-3) because these account for the majority of simulations.**